

# Changes of cellular stress response related *hsp70* and *abcb1* transcript and Hsp70 protein levels in Siberian freshwater amphipods upon exposure to cadmium chloride in the lethal concentration range

Marina V. Protopopova[1,2,*], Vasiliy V. Pavlichenko[1,2,*] and Till Luckenbach[3]

[1] Siberian Institute of Plant Physiology and Biochemistry SB RAS, Irkutsk, Russia
[2] Faculty of Biology and Soil Sciences, Irkutsk State University, Irkutsk, Russia
[3] Department of Bioanalytical Ecotoxicology, Helmholtz Centre for Environmental Research—UFZ, Leipzig, Germany
[*] These authors contributed equally to this work.

Corresponding author
Till Luckenbach,
till.luckenbach@ufz.de

## ABSTRACT

The induction of cellular stress response systems, heat shock protein *hsp70*/Hsp70 and multixenobiotic transporter *abcb1*, by cadmium chloride ($CdCl_2$) was explored in amphipod species with different stress adaptation strategies from the Lake Baikal area. Based on the lethal concentrations (LC) of $CdCl_2$, the sensitivities of the different species to $CdCl_2$ were ranked (24 hr LC50 in mg/L $CdCl_2$ (mean/95% confidence interval)): *Gammarus lacustris* (1.7/1.3–2.4) < *Eulimnogammarus cyaneus* (2.9/2.1–4.0) < *Eulimnogammarus verrucosus* (5.7/3.8–8.7) < *Eulimnogammarus vittatus* (18.1/12.4–26.6). Conjugated dienes, indicating lipid peroxidation, were significantly increased after 24 hr exposures to 5 mg/L $CdCl_2$ only in the more $CdCl_2$-sensitive species *G. lacustris* and *E. cyaneus*. Upon treatment with 0.54 to 5.8 mg/L $CdCl_2$ for 1, 6 and 24 hrs, *hsp70* transcript levels were generally more increased after the longer exposure times and in the more $CdCl_2$-sensitive species. Relating the $CdCl_2$ exposure concentrations to LCx values revealed that across the species the increases of *hsp70* transcript levels were comparatively low (up to 2.6-fold) at $CdCl_2$ concentrations ≤LC50. Relative *hsp70* transcript levels were maximally increased in *E. cyaneus* by 5 mg/L $CdCl_2$ (≙LC70) at 24 hrs (9.1-fold increase above the respective control). When *G. lacustris* was exposed to 5 mg/L $CdCl_2$ (≙LC90) for 24 hrs, the increase in *hsp70* was in comparison to *E. cyaneus* considerably less pronounced (3.0-fold increase in *hsp70* levels relative to control). Upon exposure of amphipods to 5 mg/L $CdCl_2$, increases in Hsp70 protein levels compared to untreated controls were highest in *E. cyaneus* at 1 and 6 hrs (5 mg/L $CdCl_2$ ≙ LC70) and in *E. verrucosus* at 24 hrs (5 mg/L $CdCl_2$ ≙ LC45). Thus, when the fold increases in Hsp70 protein levels in the different amphipod species were related to the respective species-specific LCx values a similar bell-shaped trend as for *hsp70* transcript levels was seen across the species. Transcript levels of *abcb1* in $CdCl_2$ exposed individuals of the different amphipod species varied up to 4.7-fold in relation to the respective controls. In contrast to *hsp70*/Hsp70, *abcb1* transcripts in $CdCl_2$ exposed individuals of the different amphipod species did not indicate similar levels of induction

of *abcb1* at equal LCx levels across the species. Induction of *hsp70* and *abcb1* genes and Hsp70 proteins by $CdCl_2$ in the lethal concentration range shows that these cellular responses are rather insensitive to $CdCl_2$ stress in the examined amphipod species. Furthermore, the increase of expression of these cellular defense systems at such high stress levels suggests that induction of these genes is not related to the maintenance of normal metabolism but to mitigation of the effects of severe toxic stress.

## INTRODUCTION

Cadmium is a non-essential heavy metal entering the environment *via* various anthropogenic and natural sources. It causes poisoning in humans and wildlife at low concentrations (*Pinot et al., 2000*). Toxic cadmium effects have often been related with increased levels of reactive oxygen species (ROS) and reactive nitrogen species (RNS) that cause damage of biological macromolecules such as proteins (*Nemmiche, 2017*). Exposure to cadmium was found to lead to an increase in the levels of transcripts of proteins encoded by the cellular stress response genes *hsp70* (*Blechinger et al., 2002*; *Da Silva Cantinha et al., 2017*; *Eufemia & Epel, 2000*; *Haap & Köhler, 2009*; *Jung & Lee, 2012*; *Kim et al., 2014*; *Lee et al., 2006*; *Mlambo et al., 2010*; *Piano, Valbonesi & Fabbri, 2004*; *Schill, Görlitz & Köhler, 2003*; *Singer, Zimmermann & Sures, 2005*; *Werner & Nagel, 1997*) and *abcb1* (*Eufemia & Epel, 2000*; *Ivanina & Sokolova, 2008*; *Zucchi et al., 2010*) in a range of aquatic organisms. Induction of the *hsp70* and *abcb1* genes by cadmium can be related to the increased abundance of damaged cellular macromolecules, such as cellular membrane fragments or misfolded proteins (*Beyersmann & Hechtenberg, 1997*; *Thévenod et al., 2000*). Increased *hsp70* and *abcb1* transcript levels are therefore seen here as indication for cellular stress caused by cadmium.

Lake Baikal in Eastern Siberia, the oldest, deepest and by volume largest lake in the world, is a biodiversity hotspot with a high degree of endemicity (*Kozhova & Izmest'eva, 1998*; *Timoshkin, 2001*). Baikal's water is generally highly pristine; however, the risk of water contamination by heavy metals is increasing. In particularly the Selenga river, the largest tributary of Lake Baikal comprising almost half of the riverine inflow into the lake, is the main source of such contaminants (*Ciesielski et al., 2016*; *Kulikova et al., 2017*).

Amphipods are a dominant taxon of the benthic communities of Lake Baikal and the more than 350 endemic species and subspecies represent 45.3% of all freshwater amphipod species of the world (*Bedulina et al., 2014*; *Takhteev, 2000*). The numerous phylogenetically closely related species featuring a range of adaptation strategies are interesting models for comparative studies (*Luckenbach, Bedulina & Timofeyev, 2015*). In the here studied amphipod species, constitutive expression levels of cellular stress response genes vary within an order of magnitude. A number of studies indicate that these different degrees of species-specific gene expression are related to differences in stress tolerance

across species. Thus, constitutive *hsp70* levels relate to the species-specific differences in thermotolerance (*Axenov-Gribanov et al., 2016*; *Bedulina et al., 2013*; *Protopopova et al., 2014*). Furthermore, different, species-dependant *hsp70* and *abcb1* gene responses to exposures to organic compounds, such as humic substances (*Protopopova et al., 2014*) and phenanthrene (*Pavlichenko et al., 2015*), were found.

In this study, we aimed to obtain an insight how sensitively cellular stress response genes *hsp70* and *abcb1* respond to heavy metal stress in amphipod species with different cellular stress response capacities. We addressed the question, at which stress levels those stress response genes are induced upon exposure of animals to $CdCl_2$. The species-specific lethal $CdCl_2$ concentrations after 24 hr exposure of animals to $CdCl_2$ (LCx 24h) served as a measure for the stress levels elicited by $CdCl_2$ at different concentrations in the different species. Quantitative polymerase chain reaction (qPCR) was used to quantify *hsp70* and *abcb1* transcript levels in RNA from tissue of amphipods upon exposure to different $CdCl_2$ concentrations for up to 24 hrs. In addition, Hsp70 protein tissue levels were determined with western blot. The study was performed with four amphipod species that differ with regard to their ecological preferences and habitats and accordingly with their physiological and cellular adaptations to environmental conditions (*Axenov-Gribanov et al., 2016*; *Bedulina et al., 2013*; *Jakob et al., 2016*). Three species, all with littoral habitats, were from the Baikal endemic *Eulinogammarus* genus. In addition, the Holarctic amphipod *Gammarus lacustris* was examined. The species occurs in waters connected to Lake Baikal but not in the areas of the lake inhabited by endemic fauna.

## MATERIALS AND METHODS

### Studied species and animal sampling

The experimental species were *Eulimnogammarus cyaneus* (Dyb., 1874), *Eulimnogammarus verrucosus* (Gerstf., 1858) and *Eulimnogammarus vittatus* (Dyb., 1874), endemic to Lake Baikal, and *Gammarus lacustris* (Sars, 1863), a common species in surface waters across the Holarctic. *Eulimnogammarus cyaneus* is most abundant along the shallow shoreline (mainly down to 1 m water depth), few individuals occur at water depths down to 20 m (*Weinberg & Kamaltynov, 1998*) at temperatures between 5−13 °C (*Takhteev, 2000*). Habitats of *E. vittatus* are at water depths of down to 30 m, its abundance peak is at 2–3 m depth (*Bazikalova, 1945*); the water temperatures in its habitat range between 9−13 °C (*Takhteev, 2000*). *Eulimnogammarus verrucosus* commonly occurs close to shore across Lake Baikal at water depths of less than 1 m to down to 10 to 15 m (*Bazikalova, 1945*) at temperatures of 5−13 °C (*Takhteev, 2000*). *Gammarus lacustris* is common in shallow, eutrophic lakes with seasonal temperature fluctuations in the Lake Baikal region at water depths of 0–7 m, but the species is not found in the open Baikal (*Bekman, 1954*). Constitutive transcript and protein levels of heat shock protein 70 (*hsp70*/Hsp70) and transcript levels of ATP binding cassette (ABC) transporter 1 (*abcb1*) indicate the cellular stress response capacities of the species in the order: *Eulimnogammarus vittatus* ≈ *Eulimnogammarus verrucosus* <<*Eulimnogammarus cyaneus* <*Gammarus lacustris* (*Protopopova et al., 2014*). Indeed, the species-specific constitutive and induced *hsp70*/Hsp70 levels correlate with
higher degrees of thermotolerance of *E. cyaneus* and *G. lacustris* compared to *E. verrucosus* (*Axenov-Gribanov et al., 2016*; *Bedulina et al., 2013*; *Jakob et al., 2016*).

    *Eulimnogammarus* specimens for the experiments were sampled at the Baikal shoreline close to Irkutsk State University's biological station at the Bolshie Koty settlement (Southern Baikal). *Gammarus lacustris* was collected from a small shallow artificial pond close to Bolshie Koty ["Lake 14", for specifications of the sampling points refer to *Protopopova et al. (2014)*]. Body lengths and weights of the animals used in the experiments were: *E. verrucosus*—20 to 25 mm, 403 ± 95 mg; *E. vittatus*—18 to 20 mm, 121 ± 16 mg; *E. cyaneus*—9 to 14 mm, 17 ± 3 mg; *G. lacustris*—14 to 18 mm, 80 ± 16 mg. Upon sampling, the animals were brought to the lab in a cold box with water from the sampling sites. The water parameters of Lake Baikal and "Lake 14" measured in another year [summer 2012, refer to *Gurkov et al. (2019)*] than the amphipod samplings for this experiment (summer 2011) can be seen as representative for the water parameters at the sampling sites at Baikal/"Lake 14" at this time of the year: Temperature—11 °C/8 °C, pH—7.5/6.8; $[O_2]$—12 mg/L/9 mg/L. Animals were acclimated to lab conditions in aerated water in 2 L tanks at 6.5−7 °C for 1 to 3 days prior to the experiments. The water used for maintaining the animals in the tanks and for the experiments was withdrawn from Lake Baikal with buckets from the pier next to the Biological Station in Bolshie Koty. Lake Baikal water instead of water from the pond "Lake 14" was also used for *G. lacustris* exposures. Thus, it was avoided that different water characteristics (minerals, organic matter) caused differences in $CdCl_2$ toxicity in the different species.

## Acute toxicities of $CdCl_2$

Acute toxicities of $CdCl_2$ to amphipods were determined in 24 hr exposure experiments. Aqueous $CdCl_2$ solutions for the experiments were set up in glass tanks with 2 L of well-aerated water along a control with clean water. The water temperature in the tanks was maintained at 6.5 to 7.0 °C during the exposures. Twenty individuals of each species were placed in each tank. Concentration series with a total of eight different $CdCl_2$ concentrations across different replicate experiments were set up. One tank was set up per $CdCl_2$ concentration and control in each replicate experiment. The $CdCl_2$ concentration ranges were 0.5 to 8 mg/L for *G. lacustris* and *E. cyaneus* and 0.625 to 30 mg/L for *E. verrucosus* and *E. vittatus*. Dead animals were removed and recorded during the experiments and all animals in all tanks were counted at the end of the exposures. Exposure experiments were repeated two to three times with different $CdCl_2$ concentrations that were chosen depending on the mortality rates determined for the already tested concentrations. Cadmium concentrations in water from a control and a $CdCl_2$ exposure were determined with atomic absorption spectroscopy (AAS) at the Analytical Chemistry Department at the UFZ. The actual Cd concentration deviated from the nominal concentration by 15% (nominal: 10 $CdCl_2$ mg/L; actual: 8.5 $CdCl_2$ mg/L). All given $CdCl_2$ concentration values were accordingly adapted (i.e., the given concentration values are 15% lower than the initial nominal concentrations).

## Measurements of conjugated diene levels

Levels of conjugated dienes were measured in amphipods upon exposure to 5 mg/L $CdCl_2$ for 1, 6 and 24 hrs and in respective controls with uncontaminated water according to *Stalnaya (1977)* with some modifications. Several entire animals were pooled to obtain 150–800 mg fresh tissue, depending on the species. The tissue was homogenized in a heptane/isopropyl alcohol mixture (1:1, v/v) using a Potter-Elvehjem tissue homogenizer. The extract was then filled in a glass tube and brought up to a volume of 4.5 ml with a heptane/isopropyl alcohol mixture. One ml of distilled water was added, the mix was vigorously shaken and incubated at 25 °C for 30 min for phase separation. Half a ml of the heptane phase was mixed with ethanol in a 1:3 ratio (v/v) and the absorbance at 233 nm was measured on a SmartSpec Plus spectrophotometer (Bio-Rad) with extraction blanks used as references. An extinction coefficient of $2.52 \times 10^4$ $M^{-1}cm^{-1}$ (*Low & Nickander, 1991*) was used to determine the amount of conjugated dienes in the solution. The data are reported as nmol $g^{-1}$ wet weight. The numbers of replicates were after 1 hr/6 hrs/24 hrs: 6/6/9 ($CdCl_2$ treatment), 6/5/6 (control) for *E. verrucosus*; 6/5/5 ($CdCl_2$ treatment), 5/5/6 (control) for *E. vittatus*; 4/5/6 ($CdCl_2$ treatment), 4/4/4 (control) for *E. vittatus*; 5/5/3 ($CdCl_2$ treatment), 5/5/5 (control) for *G. lacustris.*

## $CdCl_2$ exposures for investigating cellular stress responses

To examine cellular responses of the different amphipod species to $CdCl_2$ exposure amphipods were exposed to $CdCl_2$ concentrations resembling the species-specific LC10 and LC50 values. *Eulimnogammarus cyaneus*, *E. verrucosus*, *G. lacustris* were in addition exposed to $CdCl_2$ concentrations of 1.7 mg/L ($\hat{=}$ LC50 for *G. lacustris*) and of 5 mg/L ($\hat{=}$ LC10 for *E. vittatus*). *Eulimnogammarus vittatus* was only exposed to 5 mg/L ($\hat{=}$ LC10). Concentrations of CdCl2 in the LC10/LC50-experiment were: *E. cyaneus*—0.68/2.89 mg/L, *E. verrucosus*—0.54/5.75 mg/L, *G. lacustris*—0.59/1.7 mg/L. In parallel to all $CdCl_2$ exposures controls were kept in uncontaminated water. Exposures of amphipods to $CdCl_2$ were in aerated 2 L tanks ($n = 5$/concentration) with Baikal water at 7 °C. Specimens were sampled after 1, 6, and 24 h exposures, frozen and stored in liquid nitrogen for extraction of total protein and diene conjugates or placed in QIAzol Lysis Reagent (Qiagen, Hilden, Germany) and frozen in liquid nitrogen for RNA isolation. Control animals were kept in water not contaminated with $CdCl_2$ but otherwise at equal temperature and aeration conditions as $CdCl_2$ exposed animals. One whole specimen of *E. verrucosus* or a pool of several whole animals of the other species equalling 200–500 mg of tissue were used for the further analyses. All exposure experiments were repeated three to nine times and all samples were assessed in triplicate.

## Isolation of total RNA and cDNA synthesis

Isolation of total RNA and cDNA synthesis were performed as described in *Pavlichenko et al. (2015)*. The tissue preserved in QIAzol Lysis Reagent was homogenized with a MM400 homogenizer (Retsch, Haan, Germany) followed by a phenol-guanidine-chloroform based extraction of total RNA. To improve the separation of the organic and aqueous phases MaXtract gel (Qiagen) was used according to the manufacturer. Total RNA was

purified from the obtained aqueous phase with the miRNeasy kit with DNase treatment using a QIAcube instrument (Qiagen). One µg of total RNA was reverse-transcribed to single-stranded cDNA using Oligo(dT)18 primer (Fermentas, USA, order no. SO131) and H Minus Reverse Transcriptase (Fermentas, USA, order no. EP0452) following the manufacturer's instructions.

## Quantitative real-time PCR analysis

Gene expression levels were analysed with quantitative polymerase chain reaction (qPCR) using a StepOnePlus Real-Time PCR System (Applied Biosystems) as described previously (*Protopopova et al., 2014*). The amplification was performed in a final volume of 12.5 µl using the SensiMix SYBR Low-ROX Kit (Bioline) with each primer at 200 nM. Primer sequences for qPCR of *hsp70* and *abcb1* and housekeeping genes *β-actin, gapdh* and *ef1-a* were from *Protopopova et al. (2014)*; qPCR conditions were according to *Pavlichenko et al. (2015)* and *Protopopova et al. (2014)*: 94 °C for 4 min; 35 (for *hsp70* and references genes) and 45 cycles (for *abcb1*) with 95 °C for 15 s, 60−62 °C for 15 s and 72 °C for 15 s followed to fluorescent measurement step with 78−79 °C (close to the melting temperature of the PCR products) for 10 s. The extra step for fluorescence measurement was added to avoid detection of unspecific amplification products and primer dimers. Relative expression levels of *hsp70* and *abcb1* genes were calculated with the comparative $\Delta\Delta$Ct method (*Livak & Schmittgen, 2001*) using efficiency corrected calculation models and Best Keeper levels based on the geometric mean of the Ct values of the housekeeping genes (*Pfaffl, 2006*; *Pfaffl et al., 2004*; *Vandesompele et al., 2002*).

## Western blotting

Isolation of total protein from amphipod tissue and western blotting were performed as described in *Protopopova et al. (2014)*. Tissues were homogenized in 0.1 M Tris buffer (pH = 7.6) containing 1.5 mg phenylmethylsulfonyl fluoride (PMSF) and 1% protease inhibitor cocktail (Amresco, USA); proteins were then subjected to denaturation in Tris-HCl buffer [0.25 M Tris-HCl (pH = 6.8), 5% sodium dodecyl sulphate (SDS), 0.5 mM ethylenediamine tetraacetic acid (EDTA), 10% glycerol, 5% ß-mercaptoethanol and 0.03% Bromophenol Blue] at 95 °C for 5 min. Total protein concentrations were determined following *Lowry et al. (1951)*. Equal amounts of total protein per sample were separated by SDS gel electrophoresis on a 12.5% polyacrylamide gel (*Laemmli, 1970*) using a Mini-PROTEAN II Electrophoretic Cell (Bio-Rad, USA). Wet-transfer of the proteins from the gel to a polyvinylidene difluoride transfer membrane (GE Healthcare, UK) was according to *Towbin, Staehelin & Gordon (1979)* with minor modifications. Hsp70 protein was labelled with anti-HSP70 (Sigma-Aldrich, order no. H9776) at 1:3000, which detects both the inducible and constitutive forms of HSP70, and with secondary antibodies conjugated with alkaline phosphatase (Anti-Mouse IgG:AP Conj., Stressgen, order no. SAB-101) at 1:1000. Anti-actin antibody (Sigma-Aldrich, order no. A2668) dissolved 1:400 and secondary antibodies (Anti-Rabbit IgG, Sigma, order no. A9919) dissolved 1:1000 were used for actin labelling. Sixty ng of bovine Hsp70 (Sigma-Aldrich, order no. H9776) and 60 ng of bovine actin (Sigma-Aldrich, order no. A3653) were used as positive controls. Bands

were visualized with 0.4 mM BCIP (5-bromo-4-chloro-3-indolyl phosphate) and 0.4 mM nBT (nitroblue tetrazolium). The Gel Explorer software package (DNAtechnology, Russia) was used for semi-quantitative measurements of Hsp70 and actin levels on the membranes and Hsp70 levels were normalized relative to actin for each sample.

## Data analysis and statistics

Regressions of concentration-mortality relationships were calculated by fitting mortalities at the respective concentrations to the non-linear HILL model:

$$Y(c_{CdCl_2}) = min + \frac{(max - min)}{1 + 10^{(LogLC50 - X)*P}} \tag{1}$$

where

$\quad$ **Y** is the mortality at the $CdCl_2$ concentration $c$

$\quad$ **X** is the logarithm of the $CdCl_2$ concentration

$\quad$ **min** is the minimum percentage mortality (control, constrained to 0)

$\quad$ **max** is the maximal percentage mortality (constrained to 100)

$\quad$ **P** is the shape parameter

$\quad$ **LC50** is the $CdCl_2$ concentration causing mortality of 50% of the individuals

$\quad$ Data from qPCR, western blotting and lipid peroxidation analyses were found to satisfy the assumptions of equal variance and normality and parametric statistics were used. For pairwise comparisons of *abcb1* and *hsp70*/Hsp70 levels in treatments and respective controls at each time point the $t$-test was applied. Multiple comparisons of several treatments with one respective control were done with two-way ANOVA and Dunnett's test. To analyse the significance of observed changes in each time-point, pair-wise comparisons with the respective control were done. The results of multiple pair-wise comparisons were done using Steel-Dwass method for all pairs' comparison. Differences were regarded as significant if $P < 0.05$. Regressions were calculated with Graphpad Prism version 7. Statistical analyses were performed with JMP version 10.0 (SAS Institute, Cary, NC).

# RESULTS

## Lethal CdCl$_2$ concentrations

Acutely toxic $CdCl_2$ concentrations were across the different amphipod species from <1 to <100 mg/L (Fig. 1, Table 1). The range of LCx values calculated from $CdCl_2$ concentration—lethality relationships varied over an order of magnitude across the species. LCx values were highest for *E. vittatus* indicating the lowest sensitivity of this species to $CdCl_2$. The order of the species with regard to their sensitivities to $CdCl_2$ was: *G. lacustris* <*E. cyaneus* <*E. verrucosus* <*E. vittatus* (Fig. 1, Table 1). Below, we report the molecular effects of $CdCl_2$ on the amphipods related to the species-specific LCx values. The LCx values serve as representations of the species-specific stress levels elicited by $CdCl_2$ at different concentrations.

## Conjugated diene levels

No significant changes in conjugated diene levels ($P > 0.05$) were detected in individuals of any of the examined species exposed to 5 mg/L $CdCl_2$ for 1 and 6 hrs. After 24 hr

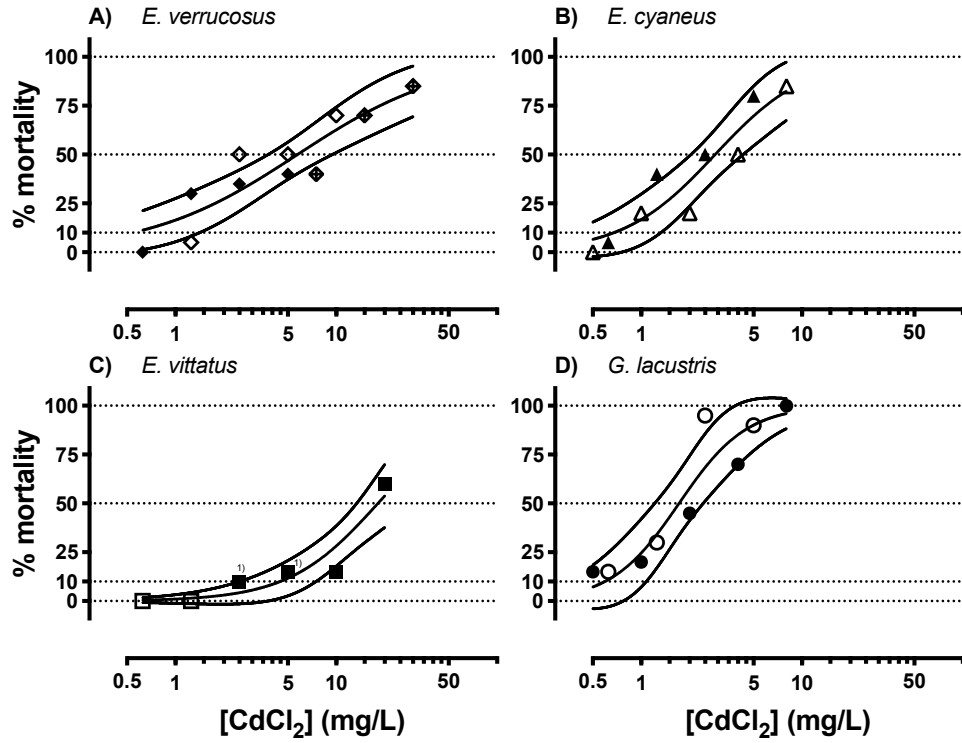

**Figure 1** **Concentration-mortality relationships for CdCl₂ in the different amphipod species.** (A) *E. verrucosus*, (B) *E. cyaneus*, (C) *E. vittatus*, (D) *G. lacustris*. Symbols represent the observed % mortality at a certain CdCl₂ concentration in a single treatment. Solid lines are regressions fitted with the HILL model (Eq. (1)) to all the data in the graph and dashed lines are the 95% confidence intervals. Each symbol stands for a single data point, symbols marked with [1] represent two data points from different experimental series. Data for % mortality were obtained in two or three separate experimental series with different CdCl₂ concentrations. Symbols represent data from a certain experimental series. Test series 1: filled symbols; test series 2: empty symbols; test series 3: symbol filled with a cross.

**Table 1** **Lethal concentration values (LCx; x: % lethality) for CdCl₂ (mg/L) and curve parameters for the different amphipod species determined with the HILL model (Eq. (1), see Fig. 1).**

|  |  | *E. verrucosus* | *E. cyaneus* | *E. vittatus* | *G. lacustris* |
|---|---|---|---|---|---|
|  | 10 | 0.54 | 0.67 | 4.47 | 0.59 |
|  | 25 | 1.76 | 1.38 | 8.98 | 1.01 |
|  | 30 | 2.31 | 1.64 | 10.63 | 1.14 |
| LCx | 45 | 4.65 | 2.52 | 16.07 | 1.55 |
|  | 50 (95% CI ) | 5.72 (3.8–8.7) | 2.89 (2.1–4.0) | 18.05 (12.4–26.6) | 1.71 (1.3–2.4) |
|  | 70 | 14.31 | 5.05 | 31.05 | 2.59 |
|  | 75 | 18.76 | 5.96 | 36.74 | 2.91 |
|  | 90 | 60.81 | 12.22 | 73.81 | 4.98 |
| Hill slope (95% CI) | | 0.20 (0.5–1.4) | 0.34 (0.7–2.3) | 0.34 (0.7–2.4) | 0.53 (0.8–3.3) |
| $R^2$ | | 0.83 | 0.87 | 0.88 | 0.88 |

**Notes.**

CI, confidence interval; $R^2$, curve fit.

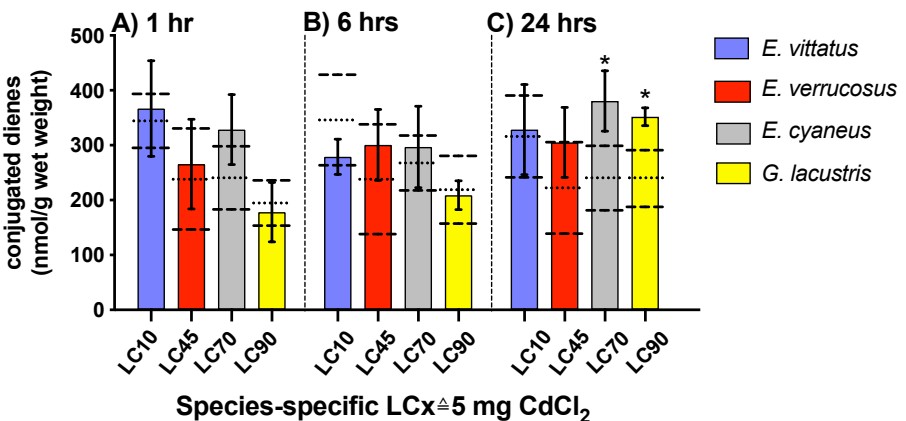

**Figure 2 Levels of conjugated dienes (CD; nmol $g^{-1}$ wet weight) in the different amphipod species upon exposure to 5 mg/L CdCl$_2$.** Exposure times to CdCl$_2$ were 1 hr (A), 6 hrs (B) and 24 hrs (C). The CdCl$_2$ concentration of 5 mg/L corresponds to the species-specific LCx values on the x-axis (refer to Table 1). Data for CdCl$_2$ exposed animals are depicted as columns (mean) and bars (standard deviations); data for the respective controls are shown as dotted (mean) and dashed (standard deviations) lines. Significant differences between treatments and respective controls are indicated by *($p < 0.05$; refer to Table 2 for *P*-values). $N = 3 - 9$ (refer to 'Materials and Methods' for details).

exposures, conjugated diene levels were significantly increased compared to the respective controls ($P < 0.05$) in *E. cyaneus* and *G. lacustris* by about 58% ($p = 0.0061$, *t*-test) and 47% ($p = 0.0018$, *t*-test), respectively (Fig. 2). The CdCl$_2$ concentration of 5 mg/L corresponds to LC90 for *G. lacustris*, LC70 for *E. cyaneus*, LC45 for *E. verrucosus* and LC10 for *E. vittatus* (Fig. 1, Table 1). Thus, the significant changes in conjugated diene levels occurred only in the more sensitive species.

## Heat shock protein 70 transcript/protein levels

*Hsp70* transcript levels across the amphipod species were upon exposures to CdCl$_2$ significantly ($p < 0.05$) increased by between 1.4—(*E. cyaneus* at LC30 for 24 hrs) to up to 9.1-fold (*E. cyaneus* at LC70 for 24 hrs) (Fig. 3, for *P*-values refer Table 2, for correspondent CdCl$_2$ concentrations to LCx values refer to Table 1). With longer exposure times to CdCl$_2$, changes in *hsp70* transcript levels were generally more pronounced in all amphipod species: After 1 hr exposures, *hsp70* levels were significantly ($p < 0.05$) increased in two cases (*E. cyaneus* at LC30, *G. lacustris* at LC90), after 6 hr exposures in four cases (*E. cyaneus* at LC30, *E. verrucosus* and *G. lacustris* at LC50, *G. lacustris* at LC90) and after 24 hr exposures in nine cases (*E. vittatus* at LC10, *G. lacustris* at LC10, LC50 and LC90, *E. verrucosus* at LC25, LC45 and LC50, *E. cyaneus* at LC30 and LC70; Fig. 3A, Table 2). Maximum *hsp70* transcript levels were at 1, 6 and 24 hrs CdCl$_2$ exposure at LC70 (*E. cyaneus*); at the lower LCx values and at LC90 the *hsp70* transcript levels were across the species lower (Fig. 3).

Upon 1 and 6 hr exposures to 5 mg/l CdCl$_2$ ($\hat{=}$ species-specific LC70), Hsp70 protein levels were significantly increased ($P < 0.05$) in *E. cyaneus* by 1.8- and 1.7-fold *vs.* the respective controls. After 24 hrs CdCl$_2$ exposure, the Hsp70 protein level showed a

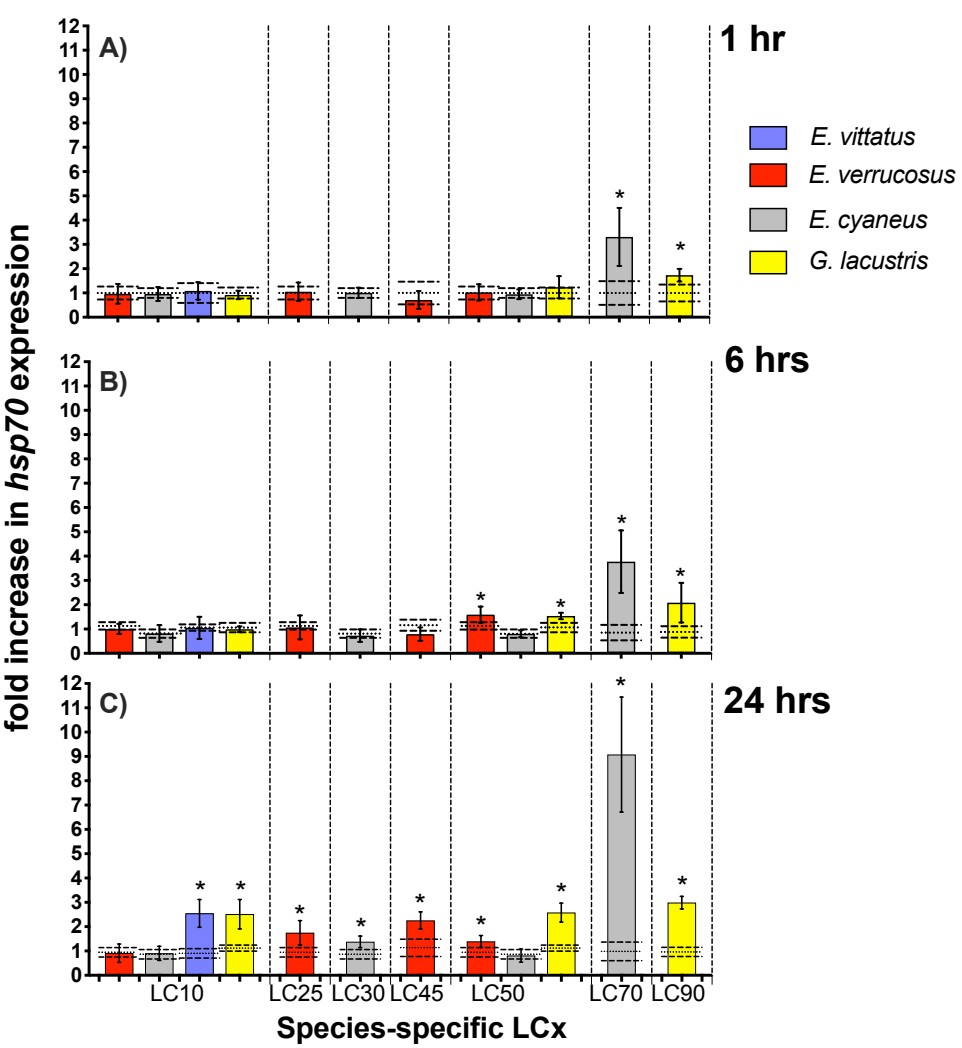

**Figure 3** *Hsp70* transcript levels in tissue of the different amphipod species upon exposure to $CdCl_2$ for 1, 6 and 24 hrs. Exposure times to $CdCl_2$ were 1 hr (A), 6 hrs (B) and 24 hrs (C). $CdCl_2$ concentrations were scaled to the species-specific LCx values (refer to Table 1 for the LCx equivalences in mg/L). Depicted are mean *hsp70* expression levels (columns) and standard deviations (bars) in $CdCl_2$ exposed animals. Expression levels in controls are shown as dotted (mean) and dashed (standard deviations) lines. Significant differences between treatments and respective controls are indicated by * ($p < 0.05$; refer to Table 2 for *P*-values).

significant increase ($P <0.05$) in *E. verrucosus* by 2.6-fold above the control (Fig. 4, Table 2; 5 mg/l $CdCl_2$ ≙ species-specific LC45). In *E. vittatus* and *G. lacustris*, Hsp70 protein levels in $CdCl_2$-exposed animals were not significantly changed ($P > 0.05$; Fig. 4). The applied $CdCl_2$ corresponded to the species-specific LC10 and LC90 values, respectively (Fig. 1, Table 1).

## *Abcb1* transcript levels

Upon exposures of amphipods to $CdCl_2$, *abcb1* transcript levels were significantly ($p < 0.05$) increased by between 2.2- (*E. cyaneus* at LC50 for 24 hrs) to up to 4.7-fold (*E. verrucosus*

**Table 2  Overview of $P$-values indicating significant differences in *hsp70*/Hsp70/*abcb1* levels between $CdCl_2$ treated amphipods and respective controls.**  Amphipods were exposed to different $CdCl_2$ concentrations expressed as lethal concentration values (LCx; x: % lethality). Respective $CdCl_2$ concentrations in mg/L are in Table 1.

| | Exposure time (h) | LC10 | LC25 | LC30 | LC45 | LC50 | LC70 | LC90 |
|---|---|---|---|---|---|---|---|---|
| *E. verrucosus* | 1 | | $P = 0.0003^{3*}$ | | | | | |
| | 6 | | | | | $P = 0.0057^1$ | | |
| | 24 | | $P = 0.0005^1$ | | $P = 0.004^1$; $P = 0.0022^2$; $P = 0.0024^3$ | $p = 0.0020^1$ | | |
| *E. cyaneus* | 1 | | | | | | $p < 0.0001^1$; $P = 0.0015^2$; $P = 0.0005^3$ | |
| | 6 | | | | | $P = 0.0003^{3*}$ | $P = 0.0002^1$; $P = 0.0104^2$; | |
| | 24 | | | $p = 0.0039^1$ | | $P = 0.0113^{3*}$ | $P < 0.0001^1$ | |
| *E. vittatus* | 1 | | | | | | | |
| | 6 | | | | | | | |
| | 24 | $P < 0.0001^1$ $P = 0.0005^3$ | | | | | | |
| *G. lacustris* | 1 | | | | | | | $P = 0.0036^1$; $P = 0.0077^3$; |
| | 6 | | | | | $p = 0.0014^1$ | | $P = 0.0082^1$; $P < 0.0001^3$ |
| | 24 | $P = 0.0008^1$ | | | | $p < 0.0001^1$ | | $P < 0.0001^1$; $P < 0.0001^3$ |

**Notes.**
[1] $P$-values indicating significant differences in *hsp70* transcript levels (depicted in Fig. 3).
[2] $P$-values indicating significant differences in Hsp70 protein levels (depicted in Fig. 4).
[3] $P$-values indicating significant differences in *abcb1* transcript levels (depicted in Fig. 5).
$P$-values marked with * were determined with Dunnet's test, all other $P$-values were determined with the $t$-test.

at LC45 for 24 hrs) (Fig. 5, Table 2). The numbers of cases of significant increases of *abcb1* transcripts differed only slightly after the different exposure times. Thus, *abcb1* transcript levels were increased in three cases after 1 hr exposures (*E. verrucosus* at LC25, *E. cyaneus* at LC70, *G. lacustris* at LC90), in two cases after 6 hr exposures (*E. cyaneus* at LC50, *G. lacustris* at LC90) and in four cases after 24 hr exposures (*E. verrucosus* at LC45, *E. cyaneus* at LC50, *G. lacustris* at LC90, *E. vittatus* at LC10) (Fig. 5, Table 2). Significant *abcb1* transcript increases at all three time points occurred only in *G. lacustris* at LC90; *abcb1* transcripts were increased at two time points in *E. cyaneus* at LC50 (Fig. 5, Table 2). In the other cases, significant *abcb1* transcript increases were only seen at one time point. Changes in *abcb1* transcript levels did not seem to depend on the $CdCl_2$ concentration or the LCx level; the significant changes were seen across different LCx values and were not generally induced at higher or lower LCx values (Fig. 5, Table 2, for LCx correspondent $CdCl_2$ concentrations values refer to Table 1).

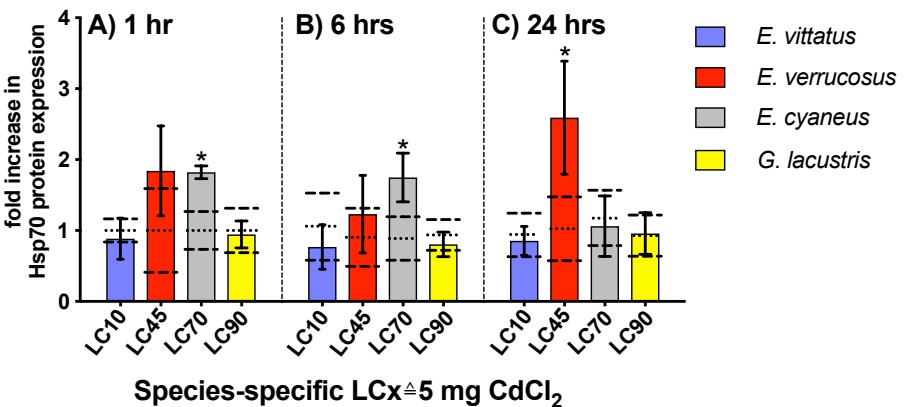

**Figure 4** **Hsp70 protein levels in tissue of the different amphipod species upon exposure to 5 mg/L CdCl₂ for 1, 6 and 24 hrs.** Exposure times to CdCl₂ were 1 hr (A), 6 hrs (B) and 24 hrs (C). CdCl₂ concentrations were scaled to the species-specific LCx values (refer to Table 1 for the LCx equivalences in mg/L). Depicted are mean *hsp 70* expression levels (columns) and standard deviations (bars) in CdCl₂ exposed animals. Expression levels in controls are shown as dotted (mean) and dashed (standard deviations) lines. Significant differences between treatments and respective controls are indicated by * ($p < 0.05$; refer to Table 2 for *P*-values).

## DISCUSSION

In this study, different amphipod species were compared regarding their sensitivities and the reactions of their cellular stress response systems to CdCl₂ exposure. The species-specific sensitivities to CdCl₂ were clearly different, as shown by LC50 values ranging over one order of magnitude (Fig. 1, Table 1). In line with sensitivities ranked by LC50 values, conjugated diene levels were only increased in *E. cyaneus* and *G. lacustris* that, based on their LCx, were the more sensitive species (Fig. 2). Conjugated dienes are primary products of lipid peroxidation resulting from oxidative stress (*Chang et al., 2011*; *Jozwik et al., 1999*), a main cause of cadmium toxicity (*Dally & Hartwig, 1997*; *Stohs & Bagchi, 1995*). The occurrence of oxidative stress caused by CdCl₂ exposure in amphipods was previously shown by the inhibition of common antioxidant enzymes in CdCl₂-exposed individuals (*Timofeyev et al., 2008*).

### Sensitivities to CdCl₂ expsoure

Experimental conditions could have influenced the sensitivities of the amphipods to CdCl₂ exposure. Thus, it is conceivable that the experimental temperature, which was equal for all species (6.5–7.0 °C), may have had an effect on the species-specific sensitivities to CdCl₂. As mentioned above in the species description ("*3.1. Studied species and animal sampling*" in the Materials and Methods section), the studied species inhabit different water zones with different temperatures. Along those lines, laboratory studies showed that the different species preferentially choose different temperature ranges, according to their habitats. Experiments, in which amphipods were kept in water with a temperature gradient, indicated that *E. verrucosus* and *E. vittatus* are adapted to colder temperatures than *E. cyaneus* and *G. lacustris*: The temperature ranges preferentially chosen were 5 °C to

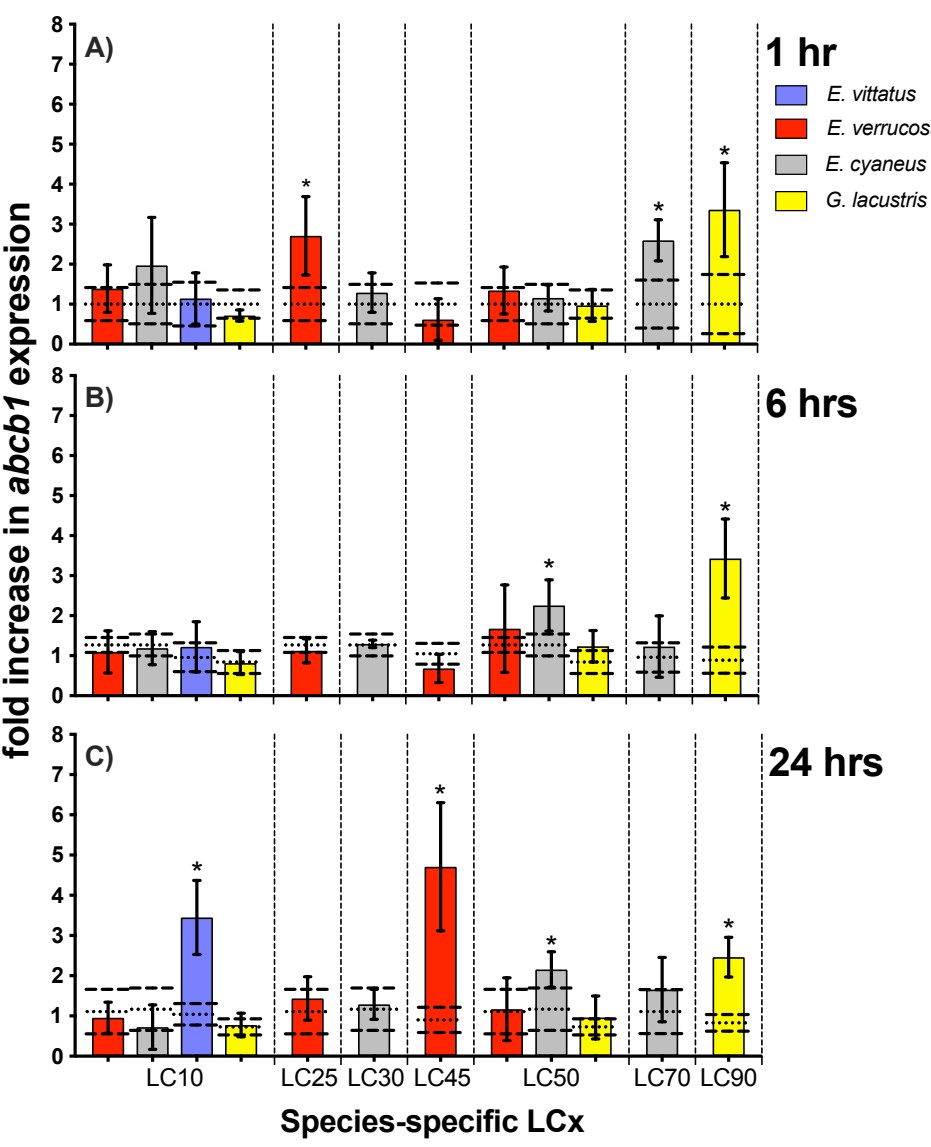

**Figure 5** Relative *abcb1* levels in amphipods upon exposure to CdCl₂ for 1, 6 and 24 hrs. Exposure times to CdCl₂ were 1 hr (A), 6 hrs (B) and 24 hrs (C). The shown *abcb1* transcript levels were scaled to the species-specific LCx values (refer to Table 1 for the LCx equivalences in mg/L). Depicted are mean *abcb1* levels (columns) and standard deviations (bars) in CdCl₂ exposed animals. Expression levels in controls are shown as dotted (mean) and dashed (standard deviations) lines. Significant differences between treatments and respective controls are indicated by *($p < 0.05$; refer to Table 2 for *P*-values).

6 °C by both *E. verrucosus* and *E. vittatus*, 11° to 12 °C by *E. cyaneus* and 15° to 16 °C by *G. lacustris* (*Timofeyev, Shatilina & Stom, 2001*; *Timofeyev & Shatilina, 2007*). It thus seems possible that *E. cyaneus* and *G. lacustris* may have been metabolically more depressed at the experimental temperature than *E. verrucosus* and *E. vittatus*. In our experiments, *E. cyaneus* and *G. lacustris* were in comparison to *E. verrucosus* and *E. vittatus* more sensitive to CdCl₂. If the experimental temperature in our study had resulted in metabolic depression in

*E. cyaneus* and *G. lacustris*, the sensitivities of those species to $CdCl_2$ may be even more increased at higher temperatures that are closer to their physiological optima.

A further parameter determining the species-specific sensitivity to $CdCl_2$ is the cadmium uptake rate into the tissue and the resulting internal cadmium levels. Cadmium uptake majorly depends on body size. It was shown with aquatic insects that the uptake rate is higher in smaller species, depending on the in relation larger body surface area across which cadmium is taken up (*Buchwalter et al., 2008*). In a study with amphipods, cadmium uptake rates during $CdCl_2$ exposures were accordingly found to be higher in the smaller and more sensitive *E. cyaneus* than in *E. verrucosus* (*Jakob et al., 2017*). The fresh weights of *E. verrucosus* specimens, as a measure of body size, were 10- to 50-fold higher than those of *E. cyaneus* specimens (*Jakob et al., 2017*); differences in fresh weights between the species were in the same range in the present study (24-fold, refer to wet weights given in section "*3.1 Studied species and animal sampling*"). *Eulimnogammarus vittatus*, however, was from the examined species least sensitive to $CdCl_2$ although the body sizes of the individuals were clearly below those of *E. verrucosus* (3.3-fold lower wetweights of *E. vittatus* compared to *E. verrucosus* specimens). It can therefore be assumed that in consequence of smaller body size the cadmium uptake rate was higher. It was previously found that the $CdCl_2$ sensitivity of amphipods may be decreased by metabolic depression: Upon exposure to $CdCl_2$, metabolic depression occurred in *E. verrucosus* but not in the more sensitive *E. cyaneus* (*Jakob et al., 2017*). Metabolic activity of the amphipods was not examined here, but it seems conceivable that *E. vittatus* may have reacted with particularly pronounced metabolic depression to $CdCl_2$ exposure, resulting in its comparatively low sensitivity despite its smaller body size compared to *E. verrucosus*.

*Gammarus lacustris* was the most sensitive species to $CdCl_2$ exposure (Fig. 1, Table 1). Strikingly, *G. lacustris* was also more sensitive than *E. cyaneus* despite considerably larger body sizes of *G. lacustris* (4.7-fold higher wet weights of *G. lacustris* than of *E. cyaneus*). Thus, for *G. lacustris* cadmium uptake rates could have been expected to be lower than for *E. cyaneus*, resulting in a consequently lower $CdCl_2$ sensitivity of *G. lacustris*. Furthermore, a metabolic depression response of *E. cyaneus* to $CdCl_2$ exposure, another potential explanation for the in comparison lower $CdCl_2$ sensitivity of *E. cyaneus*, can be ruled out: *E. cyaneus* does not react with metabolic depression to $CdCl_2$ exposure (*Jakob et al., 2017*). Furthermore, titres of metallothioneins that may be diverging in the different species can probably be excluded as reason for different sensitivities to $CdCl_2$ exposure; it was found earlier that those proteins do not seem to play an important role in cadmium detoxification in a gammarid (*Ritterhoff, Zauke & Dallinger, 1996*). It is conceivable that, independent from body size, cadmium uptake was in comparison higher in *G. lacustris* for other reasons. For instance, it was previously found that the exoskeleton of *G. lacustris* is less sturdy to mechanical pressure than that of *Eulimnogammarus* species (*Jakob et al., 2016*), indicating different compositions of the exoskeletons. Thus, because of the less rigid consistency the exoskeleton of *G. lacustris* may be better permeable for cadmium than that of the *Eulimnogammarus* species.

## Gene responses

Although metabolic depression by cadmium may have decreased activity on a physiological level in some of the species, it appeared to have had no effect on transcription activity in the amphipods. Thus, in the 6 and 24 hr CdCl$_2$ treatments at LC50 *hsp70* was significantly induced in *E. verrucosus* but not in *E. cyaneus* (Fig. 3); CdCl$_2$-caused metabolic depression occurred in *E. verrucosus* but not in *E. cyaneus* (*Jakob et al., 2017*). Furthermore, a significant *abcb1* increase was seen in *E. verrucosus* at LC25 but not in *E. cyaneus* at LC30 after 1 hr exposure to CdCl$_2$ (Fig. 5). Also *E. vittatus*, which may have been metabolically depressed by CdCl$_2$ exposure (see above), showed, as the only species, a significant increase in *abcb1* transcript levels upon a 24 hr exposure to CdCl$_2$ at LC10 (Fig. 5).

In a recent study on the whole transcriptome response of amphipods to chemical stressors, which was similarly set up as this study, it was concluded that the experimental temperature of 6 °C perturbed the transcriptome response in *G. lacustris* (*Drozdova et al., 2019b*). Thus, in comparison to *Eulimnogammarus* species, the transcriptome response in *G. lacustris* was less pronounced. Since the metabolism of *G. lacustris* appears to be less well adapted to this temperature than that of the *Eulimnogammarus* species (*Axenov-Gribanov et al., 2016*) the weaker transcriptome response in this species was seen as a result of a more decreased metabolism at 6 °C (*Drozdova et al., 2019b*). In the present study, no relation of the degree of gene responses to the species-specific adaptation to the experimental temperature became evident. If there was an effect of the temperature on the examined gene responses, it was within the observed data variation.

When relating the *hsp70* transcript or Hsp70 protein levels across the species to LCx values a bell-shaped trend of the expression data becomes obvious (Figs. 3 and 4). Such stress-level dependent expression degrees of Hsp70 have been described before: The Hsp70 titre increase in the "compensation phase" is followed by a Hsp70 titre decrease in the "non-compensation phase" (*Eckwert, Alberti & Köhler, 1997*). This Hsp70 titre decrease results from the increasingly degraded ability of the cells to react to toxicants at higher stress levels (*Eckwert, Alberti & Köhler, 1997*). The bell-shaped trend appearing in the LCx-related mRNA/protein expression data from the different species combined in the same graphs can be seen as an indication that the *hsp70* transcript/Hsp70 protein responses are corresponding in the different amphipod species at correspondent stress levels.

It was indicated earlier that across gammarid species equal internal cadmium concentrations cause toxic effects of corresponding magnitude (*Jakob et al., 2017*). Based on this finding, the LCx values may be regarded as approximate measures of the respective internal cadmium concentrations. Hence, the degrees of *hsp70* transcript and Hsp70 protein induction appear to be approximately equal at corresponding internal cadmium levels across the species.

The regulation of the *hsp70* gene and Hsp70 protein is a rapid response to stress impact on the cell (*De Nadal, Ammerer & Posas, 2011*) and has been proposed as early-warning marker for the presence of deleterious agents in the environment, affecting the organisms at an examined site (*Bierkens, 2000*; *Nadeau et al., 2001*). To be useful as a marker indicating the exposure of organisms to stress causing conditions, the *hsp70* gene/Hsp70 protein response should be sensitive, i.e., a response should occur at a sublethal stress

impact. However, the changes of *hsp70* transcript and Hsp70 protein levels at $CdCl_2$ concentrations in the lethal range in our experiments indicate that these gene/protein responses in the here examined species are rather insensitive. Concerning the stress levels, at which changes in expression levels of *hsp70* transcript/Hsp70 protein levels were found in various animal species/cell lines upon cadmium exposure, so far published studies come to diverging results: A significant increase of *hsp70* transcripts/Hsp70 protein by cadmium at levels considerably below the LC50 value was found in studies with the shrimp *Marsupenaeus japonicus* (*Ren et al., 2019*), with larvae of the midge *Chironomus tentans* (*Lee et al., 2006*), with the amphipod *Hyalella azteca* (*Werner & Nagel, 1997*) and with HeLa cells (*Ait-Aissa et al., 2000*). Changes in *hsp70* transcript / Hsp70 protein levels upon exposure to cadmium close to LC50, thus in the lethal concentration range, were seen in the redworm *Eisenia fetida* (*Brulle et al., 2006*), in the snail *Biomphalaria glabrata* (*Da Silva Cantinha et al., 2017*) and in the rotifer *Brachionus koreanus* (*Jung & Lee, 2012*). Also in the snail *Deroceras reticulatum* a Hsp70 protein increase was observed upon exposure to a rather high cadmium concentration that was above LT50 (time of expsoure lethal for 50% of the population) (*Köhler et al., 1998*). No or only slight changes of *hsp70* transcript/Hsp protein levels upon cadmium exposures below LC50 were seen in the fishes *Oreochromis mossambicus* (*Mlambo et al., 2010*) and *Acipenser persicus* (*Safari et al., 2014*). Changes in Hsp70 levels upon cadmium expsoure were also seen in the wasp *Pteromalus puparum* but effect concentrations were not linked to toxic cadmium levels (*Wang et al., 2012*).

Although our results suggest that the *hsp70*/Hsp70 responses across our examined gammarid species occur at corresponding stress levels caused by $CdCl_2$, there is no indication that the heat shock response across cellular systems/animal species is consistently evoked at similar cadmium impacts.

Across the amphipods examined here, changes in *abcb1* transcript levels upon $CdCl_2$ exposure were not uniform at corresponding LCx levels (Fig. 5, Table 2), which is in contrast to the *hsp70*/Hsp70 responses. The differences in sensitivies of the *hsp70* and *abcb1* transcript responses across the gammarid species may confirm that induction of the two genes is *via* different pathways. The induction of the *hsp70* gene is triggered by the presence of misfolded proteins that are damaged e.g., by the impact of cadmium-induced ROS (*Beyersmann & Hechtenberg, 1997*; *Thévenod et al., 2000*): According to the chaperone titration model (*Richter, Haslbeck & Buchner, 2010*), heat shock factor (Hsf1) is released when Hsps bind to the misfolded proteins thus initiating *hsp* transcription. Induction of *abcb1* appears to be triggered *via* another pathway involving the transcription factor NF-$\kappa$B (*Beyersmann & Hechtenberg, 1997*; *Thévenod et al., 2000*).

Both Hsp70 and Abcb1 proteins act as cellular protection against cadmium toxicity: the chaperone Hsp70 preserves the structure of intact proteins and re-folds damaged proteins; Abcb1, a cellular efflux transporter, appears to efflux cell membrane components, such as ceramides, which result from membrane damage by ROS and can induce apoptosis (*Beyersmann & Hechtenberg, 1997*; *Thévenod et al., 2000*). It was found earlier that *E. cyaneus*, which shows higher constitutive and induced levels of Hsp70 than *E. verrucosus* (*Bedulina et al., 2013*; *Protopopova et al., 2014*), detoxifies a higher proportion of cadmium by binding it to proteins of the biologically detoxified heat stable protein fraction (BDF)

(*Jakob et al., 2017*). The in comparison high constitutive Hsp70 titres in *Gammarus lacustris* (*Protopopova et al., 2014*) may indicate that also in this species a comparatively high level of the internal cadmium is bound to the BDF. However, when considering that the more sensitive gammarid species have higher titers of cadmium detoxifying proteins and that the transcripts of both here examined proteins are induced by cadmium only at comparatively high stress levels, it seems obvious that overall the detoxifying effect by those cellular stress response proteins on cadmium is only minor. Major determinants of the cadmium sensitivity of the species appear to be body size and in relation to that the cadmium uptake rate and one additional parameter that could be the degree of metabolic depression caused by cadmium.

Responses that could be more sensitive in the amphipods to the exposure to $CdCl_2$ than the examined genes may be the degree of lipid peroxidation and metallothionein titres (*Correi, Livingstone & Costa, 2002*). Recently, various *hsp70* isoforms were described in amphipods that could be classified as "cognate-inducible" and "inducible" (*Drozdova et al., 2019a*). As indicated by the relatively little increase in transcript level a "cognate-inducible" *hsp70* isoform was examined here: Thus, the "inducible" *hsp70* isoform may also respond more sensitively to the stress impact from $CdCl_2$ exposure.

## SUMMARY AND CONCLUSIONS

In this study, cadmium-induced *hsp70*/Hsp70 and *abcb1* levels in gammarids exposed to $CdCl_2$ were phenotypically anchored, i.e., exposure concentrations of treatments for transcript measurements were equated with the respective lethal concentrations (LCx). When gene/protein responses were related to LCx values, the four gammarid species examined showed a relatively uniform *hsp70*/Hsp70 response with pronounced responses in $CdCl_2$ treatments at higher LCx and in lower LCx treatments after longer times of exposure to $CdCl_2$. For *abcb1*, responses to $CdCl_2$ exposure, when related to LCx levels, were not uniform across the species. However, *abcb1* was, overall, also induced by $CdCl_2$ in the lethal concentration range. Induction of *hsp70*/Hsp70 and *abcb1* responses by $CdCl_2$ in the lethal concentration range in the gammarids indicates that changes in transcript levels of those genes are rather insensitive markers for cadmium stress.

### Funding

This study was financially supported by scholarships from the Deutscher Akademischer Austauschdienst (DAAD) and the Russian Ministry of Education and Science ("Mikhail Lomonosov" Programme) (Marina V. Protopopova, Vasiliy V. Pavlichenko), an international DAAD scholarship (Marina V. Protopopova, Vasiliy V. Pavlichenko), an Erasmus Mundus (MULTIC II) scholarship (Vasiliy V. Pavlichenko) and the bilateral funding programs "Helmholtz-Russia Joint Research Groups" (HRJRG) from the Helmholtz Association and the Russian Foundation for Basic Research (RFBR) (LaBeglo project HRJRG-221; Marina V. Protopopova, Vasiliy V. Pavlichenko, Till Luckenbach) and

"Helmholtz-RSF Joint Research Groups" from the Helmholtz Association and the Russian Science Foundation (RSF) (LaBeglo2/RSF 18-44-06201; Till Luckenbach). The funders had no role in study design, data collection and analysis, decision to publish, or preparation of the manuscript.

### Grant Disclosures

The following grant information was disclosed by the authors:
Deutscher Akademischer Austauschdienst (DAAD).
Russian Ministry of Education and Science.
International DAAD scholarship.
Erasmus Mundus (MULTIC II) scholarship.
The Helmholtz Association.
The Russian Foundation for Basic Research (RFBR).

### Competing Interests

The authors declare there are no competing interests.

### Author Contributions

- Marina V. Protopopova and Till Luckenbach conceived and designed the experiments, performed the experiments, analyzed the data, prepared figures and/or tables, authored or reviewed drafts of the paper, and approved the final draft.
- Vasiliy V. Pavlichenko conceived and designed the experiments, performed the experiments, authored or reviewed drafts of the paper, and approved the final draft.

### Data Availability

 Primary data and analyzed data are available in the Supplemental Files.

### Supplemental Information

Supplemental information for this article can be found online at http://dx.doi.org/10.7717/peerj.8635#supplemental-information.

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
