# Peer review of "Changes of cellular stress response related hsp70 and abcb1 transcript and Hsp70 protein levels in Siberian freshwater amphipods upon exposure to cadmium chloride in the lethal concentration range"

_PeerJ, doi:10.7717/peerj.8635_

## Round 0.1 · original submission · Major Revisions

All three reviewers agreed that your manuscript addresses a relevant topic and is suitable for PeerJ, however, in its current form it needs a major and thorough revision. The reviewers provided detailed comments which you should closely follow for the revision of your manuscript.

An important aspect is underlined by reviewer 1: Your interpretation of results is rather descriptive and lacks an in-depth discussion of the evolutionary aspects of adaptation to chemical stressors in remote and anticipated pristine areas. This may have also more general implications for the vulnerability of endemic species to pollution which should be worked-out in your discussion.

Reviewer 1 ·

Basic reporting

Remarks to Protopopova et al. : Investigation of cellullar stress response … in Siberian freshwatwer amphipods upon cadmium exposure.

This ms addresses an important aspect of stress ecology, the stress tolerance of endemic species from remote areas. The topic itself therefore warrants publication but the manuscript needs considerable revision.
The authors just describe their results in a rather simple way and solely conclude that molecular stress responses in amphipods from the Baikal follow ˈdifferent adaptation strategiesˈ and are rather ˈinsensitiveˈ. This, however is, to my opinion, just part of a much more interesting story. In fact, most of the amphipods died from Cd contamination before they could counteract stress symptoms biochemically. How come? Usually protective biochemical responses such as hsp70 induction (but also MDR-related gene activation) increase at lower concentrations than the LC50. There are many examples in the literature for this phenomenon (e.g., Brulle et al. 2006: Environ Sci Technol. 40, 2844-2850; Werner & Nagel 1997: already cited in the ms). In species from extreme habitats, however, this rule does not always apply, e.g. in ice fish which apparently do not express hsps at all. So, also in the Baikal, with its pristine water, its location in a remote area and without experiencing challenges by heat due to Siberian climate, the lack of strong selection pressure for an effective biochemical stress response system apparently has left abundant biota non-adapted to stressors, including the artificial stressor cadmium. Such argumentation would put the results of the authors in an evolutionary context and, additionally, has implications for a presumed vulnerability of these endemic species to pollution.

In the para lines 63-67, the authors simply state that, quite often, molecular effect data are generally not related to cadmium stress. This is only true, if linear or, generally, monotonous concentration-response relationships are anticipated. However, it is long known for stress proteins, including hsp70, but also other biochemical stress responses that quite often the molecular stress response collapses at concentrations lower than the LC50, due to down-regulation, pathologies etc. So, as a rule, molecular stress responses follow non-monotonous response curves with a maximum response at a given concentration and declining levels at further increasing pollutant concentration. Consequently, the statement of the authors in lines 63-67 is imprecise – already 20 years ago, the relationship between stress gene transcription, stress protein formation and demographic parameters such as mortality and reproduction has been exemplarily documented in a model snail species (Kohler et al. 1998: Environ. Toxicol. Chem. 17(11), 2246-2253).

The authors have kept and exposed all amphipods at a constant temperature of 6.5 – 7.0 °C. While this corresponds to the habitat temperatures of most tested amphipod species, it was probably too cold for E. vittatus (2 – 6°C colder that in its natural habitat). So, can the authors exclude the high mortality (line 244) in this species to result from non-physiological, low temperatures? In fact, Fig 1 does not really support the interpretation of the authors for a gradual increase in mortality in the species´ order given in lines 243 and 244, as confidence intervals for the species-specific mortality curves largely overlap, and only the one for E. vittatus differs from all the others – eventually due to the artificially low temperature which may have resulted in a metabolic depression (in contrast to the interpretation given in line 312).

A main point that necessarily has to be addressed in a revised ms is the non-justified pooling of data deriving from different LCx exposure across different species in single datasets and figures, and ˈconclusionsˈ drawn from such approach. For example, in Fig 4 each data point is defined by a specific LCx value and a – strongly covariate – specific species. It is impossible to tell LCx from species and, thus, the calculation of a regression curve on the basis of solely LCx is scientifically incorrect. The same applies to figures 2, 3, and 5. Should the significance in the data of the rightmost column in Fig 5 really be attributed to the LC90 or rather the species G. lacustris? In a revised ms, all data necessarily need to be separated according to the respective test species!

Lines 265/266: This function is definitely not Gaussian but non-monotonous with a local maximum.

Lines 360-362: The mechanistic interpretation proposed here (concomitanly high levels of hsps and metal-binding proteins) is in contrast to the results of Haap & Kohler 2009 (cited in this ms) who showed a trade-off between metal-binding proteins and hsps.

Why did the authors quantify the level of an MDR-related gene, abcb1? The MDR transporters are able to confer tolerance to organic compounds, but not cadmium. Inclusion of metallothioneins in this study would have been much more relevant in this context.

Even though the English is readable, the text, particularly in the Discussion, follows some foreign syntax and style. The authors are strongly recommended to have a revised ms pre-checked by a professional prior to submission.

Experimental design

see above

Validity of the findings

see above

Additional comments

Ms should be largely re-written. This seems to be possible but will require consideable input.

Reviewer 2 ·

Basic reporting

Please see comment below - in general the language needs some improvement. The structure, background and referencing is according to standards in the field . The results are relevant but discussion would benefit from linking the finding more with their meaning in the context fo the work or beyond.

Experimental design

Design is rather clear, with a research question being clear to me and well addressed. some more detailed comments are below:
L108 please specify the waters nutrient levels and other quality parameters.
L114ff How did the difference in Biomass for each of the species influence the bioavailability of Cd over time? 2 L is a high volume but also amphipods of a mass around 10 g would not be considered low. As this biomass varied among species, could this have any impact on the effects that one observed?

Validity of the findings

The replication seems partly relatively low but this is balanced by a repetition of the experiments. The data are provided - more detailed comments are provided in the following lines

Results
General: it is realized in some cases, but a more general consideration of effect sizes and not statistical significances is, given the heavy criticism of the latter approach, more favorable and should be prefer. Moreover exact p-values should be reported, ideally with the clear information on the replication (if possible)
L241 I assume “varied over an order of magnitude” was meant here?
L243 It is generally confusing to refer to CdCl2 as toxicant and based on this it is unclear to me whether the LCx values refer to Cd or CdCl2. I would suggest to provide information in Cd. Irrespective of this, the point needs clarification.
L276 What kind of increases is referred to here? The increases based on statistical significance? Please be specific.

Table 1 and figure 1 do largely represent the same thing – duplication of information should be avoided. Moreover, in the table, the CIs should be reported at all levels for full transparency.
Figures: I would suggest to provide instead of LC values in all the figure son the x-axes the actual concentration. The LC values are a kind of confusing and not intuitive for a reader who is not willing to read the full paper. Also the figure captions should help to explain why those concentrations have been chosen.

Discussion
The chapter would also clearly improve from some language polishing taking advantage of an expert in the field. For instance, “toxicological sensitivity” would just be “sensitivity” or “cause of toxic action” – “mode of toxic action”. Seems there are some technical terms lacking which would greatly improve readability. Moreover, it seems that some data are “just” repeated but not put into context (see L313ff but also L336f and elsewhere).
L290 Please rephrase sentence – the message is not 100% clear to me.
L297 Why only to a certain extend? Some references would be helpful as would a more clear explanation for this point.
L321f I cannot fully follow in this paragraph – I do see graphs assessing the protein levels in different species each exposed to different levels of Cd (od CdCl2) at 3 points in time. With this design, it is difficult to say something about different phases of the biological reaction as there are basically 3 points in time, which is not an overwhelming dataset and neither trend can be well support.

Additional comments

interesting work, which would benefit from the involvement of a native speaker for an English language check and also check for references and other formatting issues (see for instance L104 or 195 but also at many other occasions.).
Some other aspects are:
Abstract
The document seems to carry information on the sensitivity of amphipods towards cadmium including some gene expression data and protein production data. Although, I think I can follow the message, I would suggest to double check the language of the whole document since there are a multitude of grammatical issues (including lengthy sentences) which, if solved, would greatly increase the readability of the document for the target audience. What I found a little striking is that no information is provided on the response of Hsp70 proteins in the organisms. Also, the conclusion seems a little broad, considering that – according to the authors – only organisms from one region are used. I suggest to soften this statement.

Introduction
L34 An example of a unnecessarily lengthy sentence. It could be split in two, with the second half of the sentence representing a stand along one.
L39ff The linkage between the last sentence of the paragraph and the second last sentence is unclear to me. It should be explained why these genes are expressed and what the response entails. This might be clear for the authors but might not so for the readership in more general. Also I would suggest to limit the number of references that are cited to support one statement to a max of 3-4; exceeding this number seems not necessary to make the point, that there is support for the statement in literature.
L63ff Sadly the message of the whole paragraph is not very clear to me. Please calrify.
L68ff Although it is likely that any set of species defers in its stress response capacities, I do not see much evidence presented until now that supports this claim for the set pf species selected – there rewording and softening the statement is advised.
L76ff Also here I would suggest to revise as the data as presented until now does not support this claim. It is also recommended to add references that are in support of this statement.

Reviewer 3 ·

Basic reporting

The manuscript uses professional English; most of the used terms are similar to those in the cited bibliography.

The cited works in general help to introduce the hypothesis and to discuss this hypothesis according to the results. However, the rechecked of the cited papers such as Werner & Nagel (1997) or the inclusion of more articles can change the following sentence.

Line 63. "It is stated in many studies on the effects of cadmium on hsp70/Hsp70 levels that the test organisms were exposed to sublethal cadmium levels but the published molecular effect data are generally not related to the general level of stress caused by the cadmium concentration applied, such as lethal cadmium concentrations. It is thus unclear how sensitive the observed gene response to cadmium is."

However, Safari et al. (2014) describe the hsp70/Hsp70 genes induction using qPCR after exposure to LC50 fractions of cadmium. Also, Jung & Lee (2012) shows lethal cadmium concentration and the concentration that induced hsp70/Hsp70 gene expression also by qPCR. Mlambo et al. (2010) measured hsp70/Hsp70 proteins levels after exposure to LC05, LC10, and LC20 of three metals, including cadmium. Werner & Nagel 1997, already cited in the manuscript, shows LCx and the cadmium concentration that increased hsp70/Hsp70 proteins levels. The authors should check the findings of these manuscripts.

The manuscript has a standard structure. It describes mortality, induction of the heat shock protein hsp70/Hsp70 and the multixenobiotic transporter abcb1; and conjugated diene levels in four species of Siberian freshwater amphipods exposed to cadmium for 1, 6 and 24h.

In the abstract of the present describes a significant increase in the induction of the heat shock protein hsp70/Hsp70 and the multixenobiotic transporter abcb1 at CdCl2 concentrations similar to or higher than LC50 after 24h exposure. However, Jung & Lee 2012 and Safari et al., 2014 found induction at lower cadmium concentrations. Park & Kwak (2013) found that hsp70/Hsp70 genes were induced at very lower concentrations compared with LCx also using qPCR but after exposure to organic chemicals.

Line 25. "Induction of stress gene responses by lethal CdCl2 concentrations indicates that in the amphipods they are rather insensitive."
The pronoun "they" is OK only if the genes are insensitive.

The findings of the present manuscript could describe a particularity for amphipods, but Werner & Nagel (1997) also worked with amphipods and found an increase in hsp70/Hsp70 protein levels at very lower cadmium concentration compared with LC50

In the present manuscript, the LC50 after 24h for CdCl2 was in 1.7 mg/L in Gammarus lacustris, 2.9 mg/L in Eulimnogammarus cyaneus, 8.3 mg/L in E. verrucosus, and 18.2 mg/L in E. vittatus. However, Jakob et al. (2017) found that the LC50 after four weeks at 6°C for CdCl2 was 2 µg/L in E. cyaneus and 12.9 µg/L in E. verrucosus. Exposure duration increase sensitivity in the orders of magnitude, but this increase is species-specific. This last may help to understand the differences among the species observed in Fig. 3 over time.

Fig. 3 and Fig 4. do not show the responses for all species at the same LCx, which question the meaning of Fig. 3B and Fig 5.

References

Jakob, L., Bedulina, D. S., Axenov-Gribanov, D. V., Ginzburg, M., Shatilina, Z. M., Lubyaga, Y. A., ... & Sartoris, F. J. (2017). Uptake kinetics and subcellular compartmentalization explain lethal but not sublethal effects of cadmium in two closely related amphipod species. Environmental science & technology, 51(12), 7208-7218.

Jung, M. Y., & Lee, Y. M. (2012). Expression profiles of heat shock protein gene families in the monogonont rotifer Brachionus koreanus—exposed to copper and cadmium. Toxicology and Environmental Health Sciences, 4(4), 235-242.

Mlambo, S. S., van Vuren, J. H. J., Basson, R., & Grant, B. (2010). Accumulation of hepatic Hsp70 and plasma cortisol in Oreochromis mossambicus following sublethal metal and DDT exposure. African Journal of Aquatic Science, 35(1), 47-53.

Park, K., & Kwak, I. S. (2013). Expression of stress response HSP70 gene in Asian paddle crabs, Charybdis japonica, exposure to endocrine disrupting chemicals, bisphenol A (BPA) and 4-nonylphenol (NP). Ocean Science Journal, 48(2), 207-214.

Safari, R., Shabani, A., Ramezanpour, S., Imanpour, M. R., & Rezvani, S. (2014). Alternations of heat shock proteins (hsp70) gene expression in liver and gill of Persian sturgeon (Acipenser persicus Borodin, 1987) exposed to cadmium chloride. Iranian Journal of Fisheries Sciences, 13(4), 979-997.

Werner, I., & Nagel, R. (1997). Stress proteins HSP60 and HSP70 in three species of amphipods exposed to cadmium, diazinon, dieldrin and fluoranthene. Environmental Toxicology and Chemistry, 16(11), 2393-2403.

Experimental design

I have no questions for the experimental design or the methodology.

Validity of the findings

The hypothesis about the insensitivity of stress genes induction to cadmium exposure should be reformulated, including previous works that have already compared LCx with hsp70/Hsp70 responses.

A high level of speculation supports the discussion about a direct relationship between internal cadmium concentration and gene expression responses because the internal concentration was not measured in the present work. The debate about a relationship between size and cadmium sensitivity, should consider the size and the sensitivity of all species at the current work and include the species in others works with amphipods such as Werner & Nagel (1997).

The conclusions should take a dimension from the works that related gene stress responses and LCx previously.

Additional comments

I could not see the results of the Western-blotting, and the gene expression response for each combination between LCx and amphipod species.

---

## Round 0.2 · Minor Revisions

Reviewer 3 identified a number of minor points which should be resolved before accepting your manuscript. I am looking forward to the revised manuscript.

Reviewer 1 ·

Basic reporting

The authors did consider all of my previous comments and largely re-wrote the ms in a way that addressed them. I am fine with the new version.

Experimental design

no comment

Validity of the findings

no comment

Reviewer 3 ·

Basic reporting

General description.
This manuscript tests the hypothesis that cadmium exposure causes molecular responses in a direct form. The alternative hypothesis is that the molecular responses are an indirect consequence of cadmium effects.

The process of rejecting the hypothesis consists of comparing the lethal concentration of Cd with the concentration of Cd that induced molecular responses (effective concentration).

According to the abstract (page 1, line 30), The decision rule is the following:
An effective concentration lower than the LC50 indicates a direct response to cadmium.

Specific points

1-Figure 2.

The text in axis x is not precise and leads to a misunderstanding.

Page 8, line 157, two concentrations were tested in all species,
0 and 5 mg/L.

The x-axis does not distinguish between 5 mg/L and 0 mg/L, and among the species.
In the discussion, the authors can compare the of 5 mg/L with different LCx.

2-Figure 3
Similar to the previous figure, the text of the x-axis is not an accurate description of the experiment.

Page 8, line 172. the tested concentrations were the LC10 and the LC50; and 0, 1.7mg/L (LC50 for G. lacustris), and 5 mg/L.

3-Figure 4

Similar to the previous figures. Information is mixed with interpretations.
In the raw data archive, the tested concentrations were 0, and 5 mg/L.

The figure also does not clearly show the differences in the response of each species.
The reason for the adjustment is not apparent, because no interpolation took place.

4-Figure 5

Similar to the previous figures. For clarity, tested concentrations should be on the text of the x-axis.

The authors' interpretation can take place in the discussion.

5- Conclusion (Page 23, line 493)

Which would be the results for a not phenotiícally characteristic?

Experimental design

No comments about it

Validity of the findings

No comments about it

Additional comments

The authors obtained precise experimental results, but its description is confusing. This confusion can quickly solve by correcting the x-axis of the figures. Several previous works compare molecular and lethal responses to contaminants, only very little of this prior information was considered in the discussion. In the present manuscript, the mortality curve is monotonic, but the molecular-responses curve seems to be non-monotonic. This difference should be discussed.

---

## Round 0.3 · accepted · Accept

Thank you very much for resolving the remaining issues raised by reviewer 3.